# High-Resolution Nanotransfer Printing of Porous Crossbar Array Using Patterned Metal Molds by Extreme-Pressure Imprint Lithography

**DOI:** 10.3390/nano13162335

**Published:** 2023-08-14

**Authors:** Tae Wan Park, Young Lim Kang, Yu Na Kim, Woon Ik Park

**Affiliations:** Department of Materials Science and Engineering, Pukyong National University (PKNU), Busan 48513, Republic of Korea; twpark0125@gmail.com (T.W.P.); ylkang0914@gmail.com (Y.L.K.); kyn510000@gmail.com (Y.N.K.)

**Keywords:** metal mold, nanotransfer printing, imprint lithography, nanostructure

## Abstract

High-resolution nanotransfer printing (nTP) technologies have attracted a tremendous amount of attention due to their excellent patternability, high productivity, and cost-effectiveness. However, there is still a need to develop low-cost mold manufacturing methods, because most nTP techniques generally require the use of patterned molds fabricated by high-cost lithography technology. Here, we introduce a novel nTP strategy that uses imprinted metal molds to serve as an alternative to a Si stamp in the transfer printing process. We present a method by which to fabricate rigid surface-patterned metallic molds (Zn, Al, and Ni) based on the process of direct extreme-pressure imprint lithography (EPIL). We also demonstrate the nanoscale pattern formation of functional materials, in this case Au, TiO_2_, and GST, onto diverse surfaces of SiO_2_/Si, polished metal, and slippery glass by the versatile nTP method using the imprinted metallic molds with nanopatterns. Furthermore, we show the patterning results of nanoporous crossbar arrays on colorless polyimide (CPI) by a repeated nTP process. We expect that this combined nanopatterning method of EPIL and nTP processes will be extendable to the fabrication of various nanodevices with complex circuits based on micro/nanostructures.

## 1. Introduction

High-resolution nanofabrication methods are essential for the realization of well-arranged nanostructures with specific functionalities applicable to various advanced future device systems, such as lithium-ion batteries (LIBs) [1,2,3,4], high-performance energy harvesting devices [5,6,7], nonvolatile memories [8,9,10], solar/fuel cells [11,12,13,14], and next-generation electric vehicles (EVs) [15,16]. In particular, photolithography is currently a technology widely exploited to achieve high-level device integrations in various modern electronics [17,18,19,20]. Over the past several decades, photolithography has been important in the production of micro/nanostructures with dynamically controlled surface topographic integrated circuit (IC) structures in electronic devices [21,22]. The photolithography process is also compatible with flexible substrates, such as polyethylene terephthalate (PET) and polyimide (PI), to fabricate wearable or stretchable device systems [23,24,25]. However, despite the merits of patternability, productivity, and scalability, they have critical challenges related to their process price per unit area and the resolution limitation of approximately 50 nm. In particular, the development of a simple and new lithographic process for realizing nanoscale pattern formation at low cost is still needed.

To overcome these issues, several alternative lithography techniques, such as extreme ultraviolet (EUV) lithography [26,27], nanoimprint lithography (NIL) [28,29], dip-pen nanolithography (DPN) [30,31], atomic force microscope (AFM) lithography [32], electron-beam lithography (EBL) [33,34], and directed self-assembly (DSA) of block copolymers (BCPs) [35,36,37,38,39,40], have been consistently developed and reported for the purpose of cost-effectiveness and ultra-high-resolution pattern formation. Among these outstanding lithography techniques, nanotransfer printing (nTP) techniques have attracted much attention due to their excellent pattern resolution, patternability, and for their ability effectively to form two-dimensional (2D) and three-dimensional (3D) structures on arbitrary substrates [41,42,43,44]. On the basis of these reasons, papers on a range of unconventional transfer printing methods, including kinetically controlled nTP [45], water-assisted nTP [46], solvent-assisted nTP [47], laser-assisted nTP [48], and immersion transfer printing [49], have been continuously published. We also reported a thermally assisted nanotransfer printing (T-nTP) technique capable of producing functional nanostructures effectively over the eight-inch wafer level [50]. The fabrication of the initial mold, which is the basis of the process, is considered a very important core step in most nTP methods. However, a low-cost mold fabrication method is still required as, currently, molds manufactured by a high-cost lithography process, including photolithography, are generally used.

Herein, we introduce a new concept of the nTP process that uses surface-patterned metal master molds by extreme-pressure imprint lithography (EPIL). EPIL enables precise nanogeometries on a metallic substrate through the nanoscopic plastic deformation of the metal surface by a high load or high pressure using a Si stamp. The approach of fabricating numerous metal molds using a Si stamp can bring long-term economic feasibility in terms of the initial mold-based nTP process, which involves infinitely repeated replication steps. We also show how to fabricate various metal molds with nanopatterns by controlling the pressure based on the EPIL method. In addition, we explain how to obtain the replicated polymer patterns from the imprinted metal mold using an adhesive PI tape. Moreover, we demonstrate the reliable nanoscale pattern formation of functional materials on various surfaces via the T-nTP process.

## 2. Materials and Methods

### 2.1. Fabrication of Surface-Patterned Metal Master Molds

To create the surface nanopatterns on metallic materials, extreme-pressure imprint lithography, which can directly generate multiscale periodic micro/nanoscale pattern structures, was experimentally employed. A Si stamp fabricated by photolithography with a width/space of 250 nm was mounted onto the RAM head of press equipment capable of injecting a high load onto metal substrates. The Si mold with a depth of 250 to 350 nm used for the EPIL process was produced by conventional KrF photolithography and the reactive ion etching (RIE) process. Spin-coated positive photoresist (PR, Dongjin Semichem Co., Ltd., Republic of Korea) with a thickness of ~400 nm on the 8-inch Si wafer was exposed using a KrF scanner (Nikon, Tokyo, Japan, NSR-S203B). Then, exposed PR was developed using a developer solution (tetramethylammonium hydroxide, Dongjin Semichem Co., Ltd.). The remaining PR patterns were used as an etch mask to produce the Si pattern with plasma treatment by means of RIE (gas: CF_4_, working pressure: 7 mTorr, power: 250 W). After the removal of the residual PR areas from the dry-etched Si substrate using an RIE system, the nanopatterned Si mold with nanoscale line structures was finally obtained. All chip-level Si stamps used in this study were used by cutting an 8-inch wafer fabricated by photolithography. Zn (annealed, 99.5% metals basis, 0.25 mm, Alfa Aesar, Haverhill, MI, USA) and Al (99.99%, 0.13 mm, Sigma Aldrich Co., Ltd., St. Louis, MI, USA) substrates were directly pressed at extreme pressures of 6 and 6.5 tons, respectively, using a Si stamp fabricated by conventional photolithography. A Ni (99.9%, 0.125 mm, Sigma Aldrich Co., Ltd.) substrate was imprinted at approximately 13 tons.

### 2.2. Replication of Polymer Patterns Using Imprinted Metal Molds

During the replication step, solid poly(methyl methacrylate) (PMMA) particle with a molecular weight of 100 kg/mol (Sigma Aldrich Co., Ltd.) and liquid hydroxyl-terminated polydimethylsiloxane (PDMS-OH) with a molecular weight of 5 kg/mol (Polymer Source Inc., Dorval, QC, Canada) were used. PMMA granules were dissolved in a mixed solvent of toluene (99.5%, Junsei Co., Tokyo, Japan), acetone (99.5%, Junsei Co.), and heptane (99.0%, Junsei Co.) (*V*_tol_:*V*_ace_:*V*_hep_ = 4:4:2) to yield a 4 wt%. The solubility parameters (*δ*, MPa^1/2^) of PMMA, toluene, and acetone are 19.0, 18.3, and 19.7, respectively. Toluene and acetone are good solvents for dissolving solid PMMA. To impart a hydrophobic state on the surface of the imprinted metal substrates, the PDMS solution dissolved in toluene yielding a 2.5 wt% was spin-cast for 23 s at a speed of 5000 rpm. This step was followed by annealing at 150 °C for 2 h using a vacuum oven (OV4-30, Jeio Tech. Co., Ltd., Republic of Korea). After the spin-casting of the PMMA solution on the hydrophobic surface of the imprinted metal mold, the PMMA film was attached and detached with mild pressure (~25 kPa) using a PI adhesive tape (PIT-10050S, Isoflex, Republic of Korea). A heptane (*δ* = 15.3 MPa^1/2^), which has a poor solubility parameter for brush-treated PDMS material, can help spin-coated PMMA film to separate well from the PDMS-coated metal molds. In the replication process, a laminator system that can provide uniform pressure on the replication area was used.

### 2.3. Pattern Transfer Printing

To transfer the nanoscale pattern structures, we utilized a heat-rolling-press system (LAMIART-470 LSI, GMP Corp., Busan, Republic of Korea). This system has four rolls composed of elastic silicone rubber that can effectively provide both uniform pressure and heat. The speed of the rolls is controllable from 200 to 1500 mm/min. The rolling speed and temperature for pattern transfer printing in this study were set to 1000 mm/min and 150 °C. Deposited functional materials, in this case Au, TiO_2_, and Ge_2_Se_2_Te_5_ (GST), on PMMA replica pattern were contacted on the target substrate and passed between heated rolls. The deposition processes were implemented by physical vapor deposition (PVD) using a direct-current (DC) and radio-frequency (RF) sputtering system (DC/RF Sputtering System, KVS-T1009640, Korea Vacuum Tech. Co., Ltd., Gimpo-si, Republic of Korea). The working pressure of the main chamber was constantly maintained at 5 × 10^−3^ torr during the PVD process (applied power: 200 W). An adhesive PI tape was manually removed from the target substrate after the bonded substrate-patterning medium-PI tape passed over the rolls.

### 2.4. Characterization

The nanoscale pattern structures were observed using a field emission scanning electron microscope (FE-SEM, MIRA3, TESCAN) at an acceleration voltage of 10 kV and a working distance of 5–8 mm.

## 3. Results

### 3.1. Nanotransfer Printing Using an Imprinted Metal Mold

The process sequence for the formation of the nanoscale pattern structure using an imprinted metal mold is schematically depicted in Figure 1. This conceptual strategy has three steps: metal mold fabrication (step 1), pattern replication (step 2), and pattern transfer printing (step 3). In step 1, the metal master substrates for the core materials of the patterning process are imprinted by EPIL using a Si stamp fabricated by conventional photolithography [51]. During the EPIL process, the planar surface of the metal substrate is modified through precise plastic deformation at room temperature without the use of heat, ultraviolet (UV) light, laser, or electrical sources. In step 2, the replica pattern is produced by peeling off the polymer film from the metal master mold using an adhesive PI tape (see Section 2, Materials and Methods, for the details of the replication step). A sticky layer of PI tape (Kapton tape) helps to decouple the polymeric thin film with nanoscale surface patterns after it has been cast onto the imprinted metal mold. In step 3, nanopatterned structures can be formed on the target substrate by the T-nTP process. The functional materials are evenly deposited on the protruding surfaces of replica polymer patterns by physical vapor deposition (PVD). The functional nanoscale patterns are then transferred onto an arbitrary substrate using a heat-rolling-press system.

### 3.2. Fabrication of Surface-Patterned Metal Molds via EPIL

To produce the master molds with metallic materials, the reliable top-down hard lithography of the EPIL process was employed. Several imprinted metal molds, which can be produced using one Si stamp, serve as a viable alternative with economic value to a Si mold during the T-nTP process for the fabrication of high-resolution nanostructures. Figure 2 shows various surface-patterned metallic master molds realized by EPIL. Figure 2a displays a replicated Al substrate from an original Si stamp. An imprinted Al substrate with a width/space of 250 nm presents the reverse phase of the Si stamp. A high load or high pressure (6.5 tons) was imposed on the surface of the Al substrate during the EPIL process. Scanning electron microscope (SEM) and the inset images in the figure clearly show the replicated high-resolution nanoscale pattern structures through the plastic deformation of the Al material. Figure 2b shows the imprinted Al substrate with a width of 1 µm and space of 250 nm. This pattern structure also presents a reverse phase with the original Si stamp (line width: 250 nm, line space: 1 µm). It should be noted that precise surface patterning with different line width/space patterns can be realized by the reliable EPIL method. Figure 2c shows cross-sectional-view SEM images of a Si stamp fabricated by photolithography and various metallic substrates surface-patterned by the EPIL process. To obtain excellent pattern resolutions of Ni and Zn materials, different loads of thirteen tons and six tons were injected on the target substrates, respectively. The patterning load for the successful pattern formation of nanoscopic geometries may vary depending on the manufactured hardness based on the intrinsic characteristics of the metallic materials used.

### 3.3. Replication Using Imprinted Metal Master Molds

Figure 3 shows the replication results of a polymer material when using an imprinted metal mold. The entire process of replication is schematically illustrated in Figure 3a. Prior to pattern replication, the imprinted metal molds were pretreated with a hydroxyl-terminated polydimethylsiloxane (PDMS-OH) brush to impart a hydrophobic surface on all molds. This surface modification step facilitates the separation of the replica polymer layer from the mold. During the replication process, the coated polymer layer cannot be separated without brush treatment. We prepared three master molds (Zn, Al, and Ni) with a width/space of 250 nm via the EPIL process (Figure 3b). Photographic and SEM images of each mold indicate that the EPIL process enables nanoscale patterning on the surfaces of the metal substrates. A PMMA solution dissolved in a mixed solvent of toluene, acetone, and heptane was then spin-cast onto the surface-patterned metal master molds. Replica patterns were successfully obtained after peeling off the functional PMMA film using an adhesive PI tape. Figure 3c shows the PMMA surface patterns replicated from the metal molds. High-resolution nanoscopic line structures were successfully obtained without any defects and/or cracks compared to the surface-patterned original metal molds. These results strongly support the claim that these imprinted three metal molds can be produced from one Si stamp and can sufficiently replace a Si stamp fabricated by photolithography in terms of nanoscale pattern formation by T-nTP. Here, we should emphasize that the imprinted metal molds can be used repeatedly to duplicate PMMA patterns.

### 3.4. Formation of Functional Nanopatterns on Various Surfaces via T-nTP

Figure 4 shows the formation of periodic nanoscale patterns on various substrates via the T-nTP process using an imprinted metal master mold. Figure 4a presents transfer-printed functional line patterns with a width of 250 nm. Au lines with a width of 250 nm were reliably transferred onto the surface of SiO_2_/Si substrate. TiO_2_ and GST nanostructures were also successfully printed on the surfaces of polished metal and glass substrates, respectively. Figure 4b shows schematic illustrations of the formation of a crossbar pattern structure. The multiple transfer printing process of the T-nTP technique enables the formation of hierarchical 3D nanogeometric structures. Figure 4c shows a transfer-printed TiO_2_ nanoporous crossbar on a flexible and transparent colorless polyimide (CPI) film, indicating stable attachments even after the substrate was bent. These results clearly demonstrate that the generation of nanoscale patterns on diverse surfaces and materials can be realized via the versatile T-nTP method when using surface-patterned metal molds imprinted by EPIL.

On the basis of these results for the combined method of EPIL and nTP, the possibility of applications in device systems that require the formation of high-resolution nanostructures on the surface of various substrates was confirmed. For the low-cost combined patterning method using an imprinted metal mold to become highly competitive compared to conventional lithographic technologies, such as photolithography, nanoimprint lithography, and inkjet printing, further research on shape control of the surfaces of metal molds and selective mirror polishing of surface-patterned metal substrates should be needed. Moreover, the development of a high-throughput automatic equipment system for the combined patterning technique will improve the reliability of the findings presented in this study.

## 4. Conclusions

In summary, a novel nanopatterning method that combines the EPIL and nTP processes to obtain well-ordered functional nanostructures was proposed. We fabricated various metallic nanomolds, in this case Zn, Al, and Ni, with periodic line structures from a Si stamp using the EPIL process without the use of an imprint resist or heat, laser, UV light, or electrical sources. The surface-patterned metal substrates were utilized as a master mold that can replace the Si stamps typically used during pattern transfer printing processes. We successfully replicated polymer patterns from the imprinted metal molds, and obtained well-ordered nanoscale line patterns with different line widths using PMMA replica patterns. We also demonstrated nanoscale pattern formation with functional materials, specifically, Au, TiO_2_, and GST, on various surfaces including SiO_2_/Si, polished metal, and slippery glass via the T-nTP process. Furthermore, we undertook the pattern generation of a TiO_2_ nanoporous crossbar structure on CPI film by the multiple T-nTP technique. We expect that this approach can be applied to other emerging device fabrication processes that involve complex functional nanostructures.

## Figures and Tables

**Figure 1 nanomaterials-13-02335-f001:**
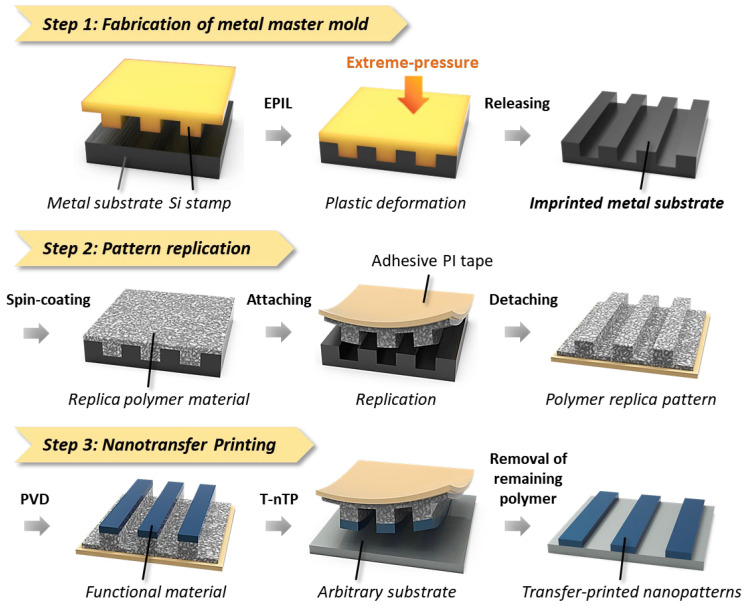
Process sequence of nanoscale pattern formation using imprinted metal molds. (Step 1: Fabrication of the metal master mold) The metal substrate is patterned by the extreme-pressure imprint lithography (EPIL) process. A metal substrate was directly patterned as a submicron-scale feature using a Si stamp fabricated by conventional photolithography. (Step 2: Pattern replication) The replica pattern is formed by attaching and detaching the spin-cast polymeric film on the imprinted metal master mold using an adhesive polyimide (PI) tape with a sticky side. (Step 3: Nanotransfer printing) The functional material is deposited onto the protruding parts of the replica pattern by the physical vapor deposition (PVD) process. The functional nanoscale patterns are then transferred onto an arbitrary substrate via the thermally assisted nanotransfer printing (T-nTP) process.

**Figure 2 nanomaterials-13-02335-f002:**
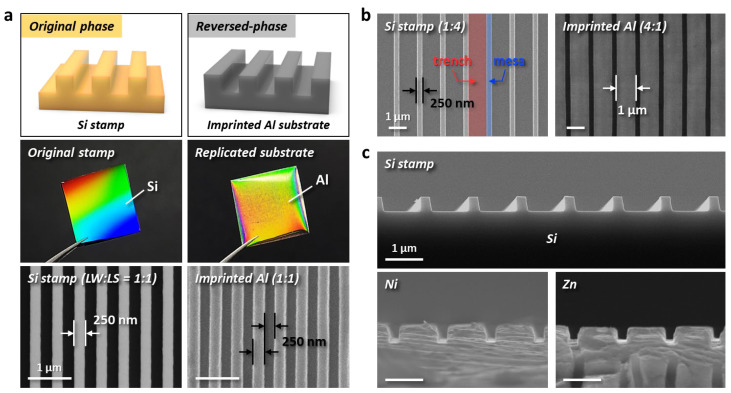
Surface-patterned metallic master molds by the direct EPIL process. (**a**) Replicated Al substrate with a width/space of 250 nm from the original Si stamp via the EPIL process. Surface-patterned Al substrate showing the reverse phase of the original Si stamp. (**b**) Imprinted Al substrate with different line-space widths of 1 µm–250 nm. (**c**) Various imprinted metal molds with Ni and Zn materials from the Si stamp. The cross-sectional-view SEM images clearly show the nanopatterned Si stamp and imprinted surfaces of Ni and Zn by EPIL. Scale bars, 1 µm (**a**–**c**).

**Figure 3 nanomaterials-13-02335-f003:**
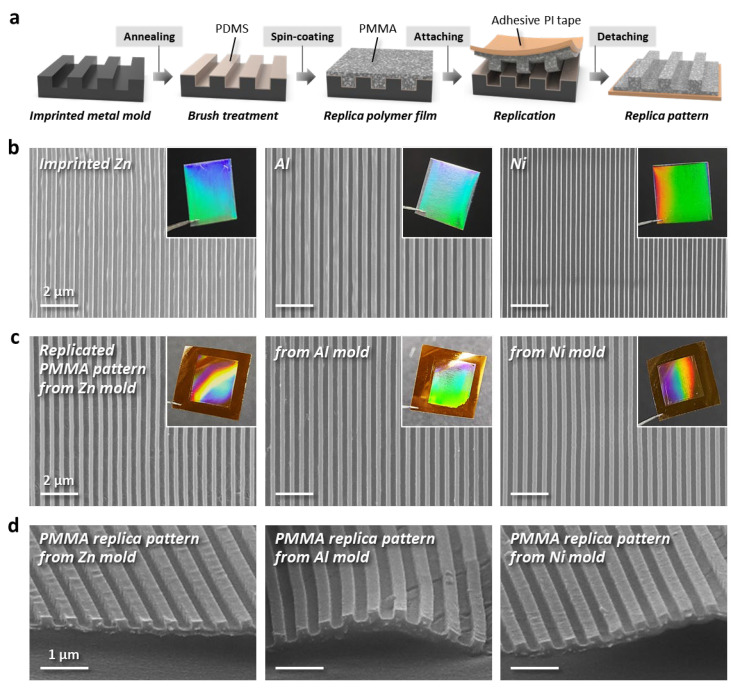
Replication of PMMA line patterns obtained from various imprinted metallic molds. (**a**) Procedure of the replication process. Before the pattern replications, the surfaces of metal molds are treated with a PDMS brush for easy separation of the replica material from the mold. (**b**) Imprinted metal (Zn, Al, and Ni) substrates via EPIL. Scale bars, 1 µm. (**c**) The replica patterns from the surface-patterned metal master molds. The insets show photographic images of the replica pattern from the imprinted metal molds. Scale bars, 1 µm. (**d**) Cross-sectional SEM images of replicated high-resolution PMMA patterns (line-width/space-width: 250 nm) from the Zn, Al, and Ni molds. Scale bars, 1 µm.

**Figure 4 nanomaterials-13-02335-f004:**
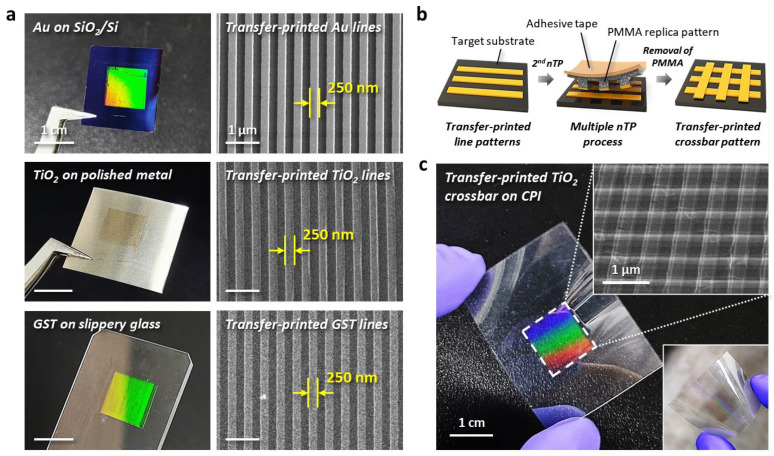
Various functional line structures and nanoporous crossbar array by the reliable nTP process. (**a**) Photographic and SEM images of line patterns with a 250 nm width printed with Au, TiO_2_, and GST materials onto SiO_2_/Si, polished metal, and slippery glass substrates, respectively. Scale bars, 1 µm. (**b**) Schematic illustration of the formation of the crossbar pattern structure. A multiple nTP process on a target substrate can effectively generate nanoscale 3D porous structures. (**c**) Transfer-printed TiO_2_ crossbar with a width of 250 nm on a colorless polyimide (CPI) substrate.

## Data Availability

The data presented in this study are available on request from the corresponding author.

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
