# Peer review of "High-Resolution Nanotransfer Printing of Porous Crossbar Array Using Patterned Metal Molds by Extreme-Pressure Imprint Lithography"

_nanomaterials, 2023, doi:10.3390/nano13162335_

Round 1

Reviewer 1 Report

In this manuscript, the authors introduced a nanotransfer printing method based on metal mold, which was fabricated by direct extreme-pressure imprint lithography with the silicon mold. This work is a beneficial attempt for nanoimprint lithography. The manuscript can be accepted for publication in Nanomaterials.

For me, this hard to hard imprint method is not recommended, even if the authors achieve good imprint results. The hard imprint under extreme high pressure will undoubtedly cause damages to the silicon mold and defects on the metal substrate. Electroplating with nanostructured polymer templates should be a much better choice for preparing metal mold.

Author Response

[General review] In this manuscript, the authors introduced a nanotransfer printing method based on metal mold, which was fabricated by direct extreme-pressure imprint lithography with the silicon mold. This work is a beneficial attempt for nanoimprint lithography. The manuscript can be accepted for publication in Nanomaterials.

For me, this hard-to-hard imprint method is not recommended, even if the authors achieve good imprint results. The hard imprint under extreme high pressure will undoubtedly cause damages to the silicon mold and defects on the metal substrate. Electroplating with nanostructured polymer templates should be a much better choice for preparing metal mold.

[Our response] We greatly appreciate the reviewer's positive and thoughtful evaluation of our research paper. We agree with the reviewer that the EPIL method can damage the Si mold and cause defects in the metallic substrate. We appreciate the reviewer’s helpful comments for the alternative choice of electroplating process for preparing the metal mold. To avoid the damages from extreme pressure, we used the duplicated Ni mold with a DLC layer fabricated by the Si mold to pattern the metallic Al substrate, as we previously reported in the journal of ACS Nano.

Figure. Method to avoid the damage of Si mold from extreme pressure. The duplicated Ni mold with a DLC layer fabricated by the Si mold can be used to pattern the metallic Al substrate. (Park et. el., ACS Nano 2021, 15, p.10464-10471)

Figure. Cross-sectional TEM images of a Ni mold without/with a 20-nm-thick DLC layer. (Park et. el., ACS Nano 2021, 15, p.10464-10471)

Reviewer 2 Report

This paper discusses a nano transfer printing technique. Starting from a nanostructure silicon master, a metal imprint is fabricated by high-pressure imprinting. This metal component is later covered by PDMS and used for casting a PMMA replica of the master. Finally another material is deposited on the nanostructure PMMA stamp and transferred on another substrate by contact printing.

The paper is original and moderately interesting, but clarifications on a number of points is required before publication:

1- The authors should be more cautious with the use of adjectives and only use the ones that are really necessary.

2- lines 26 to 40: This paragraph is too general and goes a little in every direction, please re-write it such that it is a little more focused and such that it makes sense with the specific aim of the research presented. 

3- The introduction should guide the reader from the general context to the specific need that the paper is answering. It should explain why the method developed by the authors is needed and compare it to similar methods that are already existing, underlining the differences and improvements it provides.

4- lines 65 to 72: These lines paraphrase the abstract and lines 195 to 203 also paraphrase the abstract. Readers have no interest in reading three times the same text. The abstract is a summary of the paper. The end of the introduction is not a summary, but should explain how the novelty of the method discussed in the paper is implemented, in general terms. The conclusion is not a summary either, but should discuss what has been achieved, how the research done extends the current state of the art and what can be expected from it in terms of impact.

5- line 77 "a Si stamp": Please indicate how the Si stamp was obtained. It is only written in the legend of figure 1, that the Si stamp was made by conventional photolithography. Please also explain it in detail in the text and give some details on the dimensions both of the stamp and of its patterns. After discussing the drawbacks of photolithography at length in the introduction, the authors use a silicon component made by photolithography as master. This feels like shooting yourself in your own foot. ?

6- line 89 "the PDMS solution dissolved in toluene...": What exactly is the ratio between PDMS and toluene? line 153 discusses hydroxyl-terminated PDMS, so is it PDMS or hydroxyl-terminated PDMS? 

7- paragraph 2.1, lines 112 to 127: The readers already know almost all that is explained in this paragraph, as it was discussed before in the "Materials and methods" section. Please reduce the redundancy in the text, such that the readers do not read the same thing again and again.

8-line 130 "a superior alternative": Why exactly is metal a superior alternative? Silicon seems to be a stronger material as it can be directly imprinted in metal with extreme pressure? Additionally, all further process steps can also be performed using silicon as component.

9- line 137 "the figure clearly show the replicated high resolution...": Actually, it is difficult to really have a clear idea of the process resolution from the SEM pictures. Could you please provide profilometer measurements comparing the silicon master profile and the obtained embossed metal one, such that a quantitative comparison can be performed.

10- line 142  "Figure 2c shows ... energy dispersive X-ray spectroscopy elemental mapping results": EDS is a method that is used in general when different materials are intertwined and allows to give a composition map of a sample. Here each sample contains only one metal and nothing else. So what do we learn from the EDS pictures? Probably nothing. Please remove them.

11-line 146 "the patterning load for the successful pattern formation...": Please provide a graph showing the efficacy of pattern transfer vs. applied pressure, in the case of different metals. 

12 Line 153 " to impart a hydrophilic surface.." What is the thickness of the PDMS layer? 

13-Figure 2 an and b, inserts showing Fourier transform patterns: These are too small and too dark to read. It may be better to remove them as they are impossible to read and don't convey much useful information.

14- Figure 2a, bottom panel: The imprinted Al structures show a white line on each of their sides, which seems to indicate that the replicated structures do not have sharp angles, but are rounded. Could the authors comment on that? Again profilometer measurements would be useful.

15- Figure 2c: The top corners of the structures in Ni and Zn do not seem to be very sharp. Could the authors also show the same SEM picture of the corresponding silicon stamp, such that the readers can estimate if the pattern transfer is faithful?

16-line 164 "replace a Si mould": So the metal holds would replace a silicon mould, but you need a silicon component to make the metal mould... This seems a little strange. Additionally, what you can do with metal moulds, can also be done with silicon moulds, so it is not perfectly clear what is the advantage of this process and how it can be justified.

17- Figure 3a. This figure is a little too similar to figure 1. It is not needed. 

18- Figure 3b and c as well as Figure 4a, inserts showing Fourier transforms: Again these are much too small and too dark to read. Remove them.

19-Figure 3b and c: The SEM pictures seen from the top are nice, but it would be interesting to show the obtained profiles from the sides too, as in figure 2c.

20 -line 170 "Au with a width of 250µm": What is the thickness of the transferred Au pattern?

21- Line 171 "GST" : Is GST the abbreviation for Ge2Sb2Te5? 

22 line 174 "well arranged periodic line structures": Is replication always perfect, even on large areas? Are there defects? Please provide statistics.

23 line 188: There is no discussion in the paper. Please compare the results obtained to the state of the art, compare honestly the advantages and drawbacks of the method to the advantages and drawbacks of other methods that can be used to make similar components, indicate if there are applications that are really in need of such a new fabrication method and what bottlenecks will be overcome by using it. 

English language is mostly OK, except for the excessive use of adverbs and adjectives such as outstanding, superior, advanced, high-performance, etc..

Author Response

[General review] This paper discusses a nano transfer printing technique. Starting from a nanostructure silicon master, a metal imprint is fabricated by high-pressure imprinting. This metal component is later covered by PDMS and used for casting a PMMA replica of the master. Finally, another material is deposited on the nanostructure PMMA stamp and transferred on another substrate by contact printing.

 [Our response] We greatly appreciate the reviewer’s positive and thoughtful evaluation for our research paper.

[Comment 1] The paper is original and moderately interesting, but clarifications on a number of points is required before publication: The authors should be more cautious with the use of adjectives and only use the ones that are really necessary.

 [Our response 1] We appreciate the reviewer’s helpful comment.  We corrected the use of unnecessary adjectives, as the reviewer suggested.

[Modification of the manuscript]

“However, despite the superior merits of patternability, productivity, and scalability, they still have critical challenges related to their process price per unit area and the resolution limitation of approximately 50 nm.” on line 37 of the revised manuscript.

“We also reported a practical and useful thermally assisted nanotransfer printing (T-nTP) technique capable of producing functional nanostructures effectively over the eight-inch wafer level [50].” on line 54 of the revised manuscript.

“Moreover, we demonstrate the reliable nanoscale pattern formation of functional materials on various surfaces via the versatile T-nTP process.” on line 70 of the revised manuscript.

* [Comment 2], [Comment 3], and [Comment 4] are closely related.

[Comment 2] lines 26 to 40: This paragraph is too general and goes a little in every direction, please re-write it such that it is a little more focused and such that it makes sense with the specific aim of the research presented.

[Comment 3] The introduction should guide the reader from the general context to the specific need that the paper is answering. It should explain why the method developed by the authors is needed and compare it to similar methods that are already existing, underlining the differences and improvements it provides.

[Comment 4] lines 65 to 72: These lines paraphrase the abstract and lines 195 to 203 also paraphrase the abstract. Readers have no interest in reading three times the same text. The abstract is a summary of the paper. The end of the introduction is not a summary, but should explain how the novelty of the method discussed in the paper is implemented, in general terms. The conclusion is not a summary either, but should discuss what has been achieved, how the research done extends the current state of the art and what can be expected from it in terms of impact.

[Our response 2, 3, and 4]

We appreciate the reviewer’s comment. In response to the reviewer’s comment, we revised the manuscript and added the following sentences that can show the specific meaning and aim of this study.

[Modification of the manuscript]

“In particular, the development of a simple and new lithographic process for realizing nanoscale pattern formation at low cost is still needed.” on line 40 of the revised manuscript.

“Herein, we introduce a new concept of the nTP process that uses sur-face-patterned metal master molds by extreme-pressure imprint lithography (EPIL). EPIL enables precise nanogeometries on a metallic substrate through the nanoscopic plastic deformation of the metal surface by a high load or high pressure using a Si stamp. The approach of fabricating numerous metal molds using a Si stamp can bring long-term economic feasibility in terms of the initial mold-based nTP process, which involves infinitely repeated replication steps. We also show how to fabricate various metal molds with nanopatterns by controlling the pressure based on the EPIL method. In addition, we explain to obtain the replicated polymer patterns from the imprinted metal mold using an adhesive PI tape. Moreover, we demonstrate the reliable nanoscale pattern formation of functional materials, in this case Au, TiO2, and GST, on various surfaces of SiO2/Si, polished metal, slippery glass, and colorless polyimide (CPI) substrates via the versatile T-nTP process. We expect that the newly proposed concept will be applied to other micro-to-nanoscale manufacturing processes and to complex device fabrication methods.” on line 61 of the revised manuscript.

* [Comment 5] and [Comment 16] are closely related.

[Comment 5] line 77 "a Si stamp": Please indicate how the Si stamp was obtained. It is only written in the legend of figure 1, that the Si stamp was made by conventional photolithography. Please also explain it in detail in the text and give some details on the dimensions both of the stamp and of its patterns. After discussing the drawbacks of photolithography at length in the introduction, the authors use a silicon component made by photolithography as master. This feels like shooting yourself in your own foot. ?

[Comment 16] line 164 "replace a Si mould": So the metal holds would replace a silicon mould, but you need a silicon component to make the metal mould... This seems a little strange. Additionally, what you can do with metal moulds, can also be done with silicon moulds, so it is not perfectly clear what is the advantage of this process and how it can be justified.

[Our response 5 and 16] We appreciate the reviewer’s helpful comments. This study shows both how to make patterned metal molds and how to use the molds. Although Si master mold with patterns fabricated by photolithography is used to pattern the metal substrates, the patterned metallic molds by EPIL themselves have functionality and have many applications, such as nano-channel heat sink, holographic film, current collector of battery, electrode of triboelectric device, and imprint mold for patterning other ductile metals. Among them, this paper particularly emphasizes the use as a mold in the nTP process. Despite the importance of development of facile and low-cost mold fabrication technology in various nTP methods, Si molds are generally fabricated by high-cost photolithography. Furthermore, during the EPIL process, Si mold can easily be damaged by extreme pressure. To avoid the damages from the high pressure for large-area patterning, we use an imprinted Ni mold with a DLC layer fabricated by the Si mold to pattern the metallic substrate, as we previously reported in the journal of ACS Nano. We agree with the reviewer that we did not explain why a Si mold should be used to replace the Si mold in the EPIL process. In response to the reviewer’s comment, we revised our manuscript as follows.

Figure. Method to avoid the damage of Si mold from extreme pressure for large-area patterning. The duplicated Ni mold with a DLC layer fabricated by the Si mold can be used to pattern the metallic Al substrate. (Park et. el., ACS Nano 2021, 15, p.10464-10471)

Figure. Cross-sectional TEM images of a Ni mold without/with a 20-nm-thick DLC layer. (Park et. el., ACS Nano 2021, 15, p.10464-10471)

[Modification of the manuscript]

“A Si stamp fabricated by photolithography with a width/space of 250 nm was mounted onto the RAM head of press equipment capable of injecting a high load onto metal substrates.” on line 77 of the revised manuscript.

“A metal substrate was directly patterned on the submicron scale using a Si stamp fabricated by conventional photolithography.” on caption of Figure 1.

“These results strongly support the claim that these imprinted three metal molds can be produced from one Si stamp and can sufficiently replace a Si stamp fabricated by photolithography in terms of nanoscale pattern formation by T-nTP.” on line 202 of the revised manuscript.

[Comment 6] line 89 "the PDMS solution dissolved in toluene...": What exactly is the ratio between PDMS and toluene? line 153 discusses hydroxyl-terminated PDMS, so is it PDMS or hydroxyl-terminated PDMS?

[Our response 6] We appreciate the reviewer’s comments. The PDMS that uses in this study is hydroxyl-terminated PDMS with a molecular weight of 5 kg/mol. In response to the reviewer’s comment, we revised our manuscript for surface modification of the mold by OH-PDMS as follows.

[Modification of the manuscript]

“During the replication step, solid poly(methyl methacrylate) (PMMA) particle with a molecular weight of 100 kg/mol (Sigma Aldrich Co. Ltd.) and liquid hydroxyl-terminated polydimethylsiloxane (PDMS-OH) with a molecular weight of 5 kg/mol (Polymer Source Inc.) were used.” on line 96 of the revised manuscript.

“To impart a hydrophobic state on the surface of the imprinted metal substrates, the PDMS solution dissolved in toluene yielding a 2.5 wt% was spin-casted for 23 seconds at a speed of 5,000 rpm.” on line 104 of the revised manuscript.

[Comment 7] paragraph 2.1, lines 112 to 127: The readers already know almost all that is explained in this paragraph, as it was discussed before in the "Materials and methods" section. Please reduce the redundancy in the text, such that the readers do not read the same thing again and again.

[Our response 7] We appreciate the reviewer’s kind comment. As the reviewer suggested, we revised and deleted repetitive texts in paragraph 3.1. Nanotransfer printing using an imprinted metal mold, and we also added the further experimental data and supplementations including the pioneering references, as follows.

[Modification of the manuscript]

3.1. Nanotransfer printing using an imprinted metal mold

The process sequence for the formation of the nanoscale pattern structure using an imprinted metal molds is schematically depicted in Figure 1. This conceptual strategy has three steps: metal mold fabrication (step 1), pattern replication (step 2), and pattern transfer printing (step 3). In step 1, the metal master substrates for the core materials of the patterning process are imprinted by EPIL using a Si stamp fabricated by conventional photolithography [51]. During the EPIL process, the planar surface of the metal substrate is modified through precise plastic deformation at room temperature without the use of heat, ultraviolet (UV) light, laser, or electrical sources. In step 2, the replica pattern is produced by attaching and detaching peeling off the spin-casted polymer film from the metal master mold using an adhesive PI tape (See Materials and Methods for the details of replication step). A sticky layer of PI tape (Kapton tape) helps to decouple the polymeric thin film with nanoscale surface patterns after it has been cast onto the imprinted metal mold. In step 3, nanoscale patterned structures can be formed on the target substrate by the T-nTP process. The functional materials are evenly deposited on the protruding surfaces of replica polymer patterns by physical vapor deposition (PVD). The functional nanoscale patterns are then transferred onto an arbitrary substrate using a heat-rolling-press system that can provide uniform heat and pressure over the patterning area.” on line XX of the revised manuscript.

“A Si stamp fabricated by photolithography with a width/space of 250 nm was mounted onto the RAM head of press equipment capable of injecting a high load onto metal substrates. The Si mold with a depth of 250 to 350 nm used for the EPIL process was produced by conventional KrF photolithography and reactive ion etching (RIE) process. Spin-coated positive photoresist (PR, Dongjin Semichem Co. Ltd.) with a thickness of ~ 400 nm on the 8-inch Si wafer was exposed using a KrF scanner (Nikon, NSR-S203B). Then, exposed PR was developed using a developer solution (tetramethylammonium hydroxide, Dongjin Semichem Co. Ltd.). The remaining PR patterns were used as an etch mask to produce the Si pattern with plasma treatment by means of RIE (gas: CF4, working pressure: 7 mTorr, power: 250 W). After the removal of the residual PR areas from the dry-etched Si substrate using an RIE system, the nano-patterned Si mold with nanoscale line structures was finally obtained. All chip level Si stamps used in this study were used by cutting an 8-inch wafer fabricated by photolithography. Zn (annealed, 99.5% metals basis, 0.25 mm, Alfa Aesar) and Al (99.99%, 0.13 mm, Sigma Aldrich Co. Ltd.) substrates were directly pressed at extreme pressures of 6 and 6.5 tons, respectively, using a Si stamp fabricated by conventional photolithography. A Ni (99.9%, 0.125 mm, Sigma Aldrich Co. Ltd.) substrate was imprinted at approximately 13 tons.” on line 77 of the revised manuscript.

“Figure 1. Process sequence of nanoscale pattern formation using imprinted metal molds. (Step 1: Fabrication of the metal master mold) The metal substrate is patterned by the extreme pressure imprint lithography (EPIL) process. A metal substrate was directly patterned on the submicron scale using a Si stamp fabricated by conventional photolithography” on line XX of the revised manuscript.

[Comment 8] line 130 "a superior alternative": Why exactly is metal a superior alternative? Silicon seems to be a stronger material as it can be directly imprinted in metal with extreme pressure? Additionally, all further process steps can also be performed using silicon as component.

[Our response 8] We appreciate the reviewer’s comment. We partially agree with the reviewer that Si is a stronger material than metal. However, ductile and rigid metallic materials are more advantageous for large-area patterning in the EPIL process than Si, because Si can be easily damaged due to its brittle nature. Furthermore, in the replication process of nTP method, we can use metal molds with the pressure-controlled depths by one Si mold. In response to the reviewer’s comment, we revised our manuscript as follows.

Figure. Controllability of patterning depth depending on the pressing force and time. (a) Time-dependency of patterning depth at a fixed pressing force of 1450 kgf. (b) Load-dependency of patterning depth at a fixed pressing time of two seconds. (c) 2D graphs for time- and force-dependent patterning depth of copper film controlled by hard silicon mold with a pattern depth of 350 nm. (Park et. el., ACS Nano 2021, 15, p.10464-10471)

[Modification of the manuscript]

“Several imprinted metal molds, which can be produced using one Si stamp, serve as a viable alternative with economic value to a Si mold during the T-nTP process for the fabrication of high-resolution nanostructures.” on line 166 of the revised manuscript.

* [Comment 9] and [Comment 14] are closely related.

[Comment 9] line 137 "the figure clearly show the replicated high resolution...": Actually, it is difficult to really have a clear idea of the process resolution from the SEM pictures. Could you please provide profilometer measurements comparing the silicon master profile and the obtained embossed metal one, such that a quantitative comparison can be performed.

[Comment 14] Figure 2a, bottom panel: The imprinted Al structures show a white line on each of their sides, which seems to indicate that the replicated structures do not have sharp angles, but are rounded. Could the authors comment on that? Again, profilometer measurements would be useful.

[Our response 9 and 14] We appreciate the reviewer’s comments. As the reviewer knows, in case of the surface-patterned materials with a depth of hundreds of nanometers, the edge sides of patterns may be brightly observed under an electron beam-based microscope. In particular, this phenomenon appears more clearly on metal surfaces due to the reflective property of the material surface. For another reason, the slope of the line structure sometimes appears bright in SEM observation.

Figure. Surface-patterned various substrates. Imprinted (a) Ag (Nanotechnology, 22, 155302, 2011) and (b) Ti (Science, 346, 6215, 2014) substrate. (c) Square-shaped Si mold with a depth of 350 nm fabricated by conventional photolithography (ACS Nano, 15(6), 10464, 2021).

If excessive white lines are observed on the edge side of the pattern, it may have been derived from a study such as the one below.

Figure. Transfer-printed Pt line patterns by nTP process. (a) Procedure for the formation of L-shaped pattern by controlling the deposition angle. (b) The L-shaped Pt lines fabricated by tilt deposition with an angle of 135o. (Korean J. Met. Mater., 58(2), 145, 2020)

In response to the reviewer’s comment, we changed the SEM image with reduced reflections on the edge sides of the imprinted Al substrate using the same specimen by adjusting the image contrast in the view mode of SEM.

[Modification of the manuscript]

“Figure 2. Surface-patterned metallic master molds by the EPIL process. (a) Replicated Al substrate with a width/space of 250 nm from the original Si stamp via EPIL process. Surface-patterned Al substrate showing the reverse phase of the Si stamp. (b) Imprinted Al substrate with different line-space widths of 1 µm-250 nm. (c) Various imprinted metal molds with Ni and Zn materials from the Si stamp. The cross-section SEM images clearly show the nanopatterned Si stamp and imprinted sur-faces of Ni and Zn by EPIL.” on line 185 of the revised manuscript.

* [Comment 10] and [Comment 15] are closely related.

[Comment 10] line 142 “Figure 2c shows ... energy dispersive X-ray spectroscopy elemental mapping results": EDS is a method that is used in general when different materials are intertwined and allows to give a composition map of a sample. Here each sample contains only one metal and nothing else. So, what do we learn from the EDS pictures? Probably nothing. Please remove them.

[Comment 15] Figure 2c: The top corners of the structures in Ni and Zn do not seem to be very sharp. Could the authors also show the same SEM picture of the corresponding silicon stamp, such that the readers can estimate if the pattern transfer is faithful?

[Our response 10 and 15] We appreciate the reviewer’s comment. We showed that the well-defined pattern was successfully imprinted from the Si stamp by adding the SEM image of the Si stamp, as shown in revised Figure 2c. We also removed EDS images, as the reviewer suggested.

[Modification of the manuscript]

“Figure 2c shows cross-section view SEM and energy-dispersive X-ray spectroscopy (EDS) elemental mapping results images of Si stamp fabricated by photolithography and various metallic substrates surface-patterned by the EPIL process.” on line 179 of the revised manuscript.

“Figure 2. Surface-patterned metallic master molds by the EPIL process. (a) Replicated Al substrate with a width/space of 250 nm from the original Si stamp via EPIL process. Surface-patterned Al substrate showing the reverse phase of the Si stamp. (b) Imprinted Al substrate with different line-space widths of 1 µm-250 nm. (c) Various imprinted metal molds with Ni and Zn materials from the Si stamp. The cross-section SEM images clearly show the nanopatterned Si stamp and imprinted sur-faces of Ni and Zn by EPIL.” on line 185 of the revised manuscript.

[Comment 11] line 146 "the patterning load for the successful pattern formation...": Please provide a graph showing the efficacy of pattern transfer vs. applied pressure, in the case of different metals.

[Our response 11] We appreciate the reviewer’s comment. In this study, we manually implemented the EPIL process using a press machine with a digital gauge of injected load. So, unfortunately, we cannot provide a graph showing applied pressure now. To make possible the reviewer’s concerns, we are developing an automatic EPIL system that can derive graphs of applied force and time during the EPIL process.

Figure. Manual EPIL system that can provide high load (~ 30 tons).

Figure. Automatic EPIL system consisting of programming unit, patterning unit, and monitoring unit. (Unpublished work)

[Comment 12] Line 153 " to impart a hydrophilic surface.." What is the thickness of the PDMS layer?

[Our response 12] We appreciate the reviewer’s comment. To obtain a hydrophobic surface of the mold, hydroxyl-terminated PDMS is often used because the CH3 in PDMS provides hydrophobicity to the surface, making it easier to separate the PMMA film from the mold. After PDMS solution dissolved in heptane is spin-coated on the mold, the mold is annealed at 130oC for 1-2 hours in a vacuum oven, followed by washed out with heptane. The thickness of the PDMS layer is estimated to be approximately 2 nm, and is very thin, making it very difficult to measure even by TEM. If PS-b-PDMS block copolymer is spin-coated and annealed, the PDMS brush layer can be observed with difficulty as follows.

Figure 2. Cross-sectional TEM images of self-assembled PS–PDMS di-block copolymer thin films depending on the brush, morphology, and film thickness. The block copolymers were self-assembled on a PS brush (a, c, e) or a PDMS brush (b, d, f). Lamella-forming BCPs (a, b, c, d) and sphere-forming BCPs (e, f, g, h) were used. For the lamellae, the film thickness (t) was also varied. (a) t = 14 nm, (b) 22 nm, (c) 40 nm, and (d) 54 nm. EDS maps of (g) Si and (h) O for the spherical morphology formed on a PS brush. For all of the samples, PDMS top layers with a thickness of 2–3 nm were observed. (Park et. al., Nano Letters 11 (10), p.4095-4101)

* [Comment 13] and [Comment 18] are closely related.

[Comment 13] Figure 2 an and b, inserts showing Fourier transform patterns: These are too small and too dark to read. It may be better to remove them as they are impossible to read and don't convey much useful information.

[Comment 18] Figure 3b and c as well as Figure 4a, inserts showing Fourier transforms: Again, these are much too small and too dark to read. Remove them.

[Our response 13 and 18] We appreciate the reviewer’s comments and agree with the reviewer’s concerns regarding the less clarity of patterns. The FFT pattern is useful data as a supplementary material of well-ordered pattern structures. In response to the reviewer’s comments, we revised FFT patterns to be clearer and bigger for the readers to easily perceive and read.

[Modification of the manuscript]

on line 185 of the revised manuscript.

on line 205 of the revised manuscript.

on line 221 of the revised manuscript.

[Comment 17] Figure 3a. This figure is a little too similar to figure 1. It is not needed.

[Our response 17] We appreciate the reviewer’s comment. Figure 3 focuses on the replication step in the nTP process, and Figure 1 shows the conceptual strategy for the whole process of this study. Therefore, we wanted to show the details of the replication step in Figure 3a. In response to the reviewer’s comment, we added a following sentence for readers to fully understand the importance of brush treatment in the replication step.

[Modification of the manuscript]

“During the replication process, the coated polymer layer cannot be separated without brush treatment.” on line 192 of the revised manuscript.

[Comment 19] Figure 3b and c: The SEM pictures seen from the top are nice, but it would be interesting to show the obtained profiles from the sides too, as in figure 2c.

[Our response 19] We appreciate and agree with the reviewer’s helpful comment. In response to the reviewer’s comment, we added the cross-section image of replicated PMMA patterns in Figure 3d.

[Modification of the manuscript]

on line 205 in the Figure 3d of the revised manuscript.

[Comment 20] line 170 "Au with a width of 250µm": What is the thickness of the transferred Au pattern?

[Our response 20] We appreciate the reviewer’s comment. Controlling the thickness of transfer-printed functional material is a very important issue in various nTP studies. The thickness of printed Au, TiO2, and GST in this study is approximately ~ 15 nm. In fact, the nTP process has a limitation in terms of the height of patterning material, as the reviewer mentioned. So, combined patterning methods or multiple patterning techniques are fundamentally required to improve the thickness of patterning medium. We are developing a new patterning strategy as below to overcome this issue.

Figure. Results for the combined patterning method of hard imprinting and nTP. (a) Sequential process of combined patterning method of imprinting and transfer printing. (b) After hard imprinting of deposited Au material with a thickness of 50 nm. (c) Transfer-printed 50 nm-thick Au patterns by nTP. (Unpublished work)

[Comment 21] Line 171 "GST" : Is GST the abbreviation for Ge2Sb2Te5?

[Our response 21] We appreciate the reviewer’s comment. The GST mentioned in line 171 is an abbreviation of Ge2Sb2Te5, as the reviewer noted. In response to the reviewer’s comment, we clarified this for the readers to fully understand.

[Modification of the manuscript]

“Deposited functional materials, in this case of Au, TiO2, and Ge2Se2Te5 (GST), on PMMA replica pattern were contacted on the target substrate and passed between heated rolls” on line 120 of the revised manuscript.

[Comment 22] line 174 "well arranged periodic line structures": Is replication always perfect, even on large areas? Are there defects? Please provide statistics.

[Our response 22] We appreciate the reviewer’s thoughtful comment. In this study, the replication process was manually performed by the hand of the experimenter because it was confined to the chip level. Nevertheless, nanoscale pattern structures are statistically well-formed within the successfully obtained replica patterns. Whereas, in the case of large-area replication over the wafer scale, a rolling-press system must be used as shown below because various defects frequently occur with a very high probability.

Figure. Replication of nanopatterns at an 8-inch wafer scale. (a) A rolling-press system that can provide uniform pressure. (b) Photographic images after attaching (left), during the detachment process (middle), and after detaching (right) during the replication step, all dome by hand. (c) Top- and tilt-view SEM images of (b), showing defective replica patterns with distortions and many microcracks. (d) photographs taken after attaching (left), during detachment step (middle), and after detaching (right) in the replication step using a rolling-press system. (e) Top- and tilt-view SEM images of (d), showing well-defined PMMA line/space patterns. (Park et.el., Sci. Adv., 6, eabb6462, 2020)

[Comment 23] line 188: There is no discussion in the paper. Please compare the results obtained to the state of the art, compare honestly the advantages and drawbacks of the method to the advantages and drawbacks of other methods that can be used to make similar components, indicate if there are applications that are really in need of such a new fabrication method and what bottlenecks will be overcome by using it.

[Our response 23] We appreciate the reviewer’s comment and agree with the concern regarding the absence of discussion in the manuscript. In response to the reviewer’s comment, we added the discussion suggesting the shortcoming and future direction of this study.

[Modification of the manuscript]

“On the basis of these results for the combined method of EPIL and nTP, the possibility of applications in device systems that require the formation of high-resolution nanostructures on the surface of various substrates was confirmed. For the low-cost combined patterning method using an imprinted metal mold to become highly competitive compared to conventional lithographic technologies, such as photolithography, nanoimprint lithography, and inkjet printing, further research on shape control of the surfaces of metal molds and selective mirror polishing of surface-patterned metal substrates should be needed. Moreover, the development of a high-throughput automatic equipment system for the combined patterning technique will improve the reliability of the findings presented in this study.” on line 229 of the revised manuscript.

Reviewer 3 Report

Good paper both in presentation and content, should be accepted after minor.

The only disadvantage is that the statistic data analysis is lacking.

Author Response

[General review] Good paper both in presentation and content, should be accepted after minor. The only disadvantage is that the statistic data analysis is lacking.

[Our response] We really appreciate the reviewer’s positive evaluation of our research paper. This study focuses on the formation of nanoscale patterns on various surfaces of substrates based on the development of a novel combination patterning strategy of extreme pressure imprint lithography and thermally assisted nanotransfer printing. We successfully obtained line and crossbar structures with a width of 250 nm on the diverse substrates. We also agree with the reviewer that further statistical data and analysis are needed. In response to the reviewer’s helpful suggestion, we have added experimental and analytical supplements to provide a clearer statistical picture of the details of pattern formation during the patterning process, as follows.

[Modification of the manuscript]

on line 185 in the Figure 2c of the revised manuscript.

on line 205 in the Figure 3d of the revised manuscript.

Round 2

Reviewer 2 Report

The authors have implemented a number of modifications in the text, which improved the manuscript.

Some points still remain unresolved:

1- Figure 2, 3 and 4, Fourier transform patterns: These are still too small and too dark to read. Please rearrange the elements in these figures such that the Fourier transforms are larger and the reader can see the information these Fourier transforms convey.

2- Fourier transform patterns: These Fourier transform patterns are not discussed in the main text. In fact, they are shown in Figure 2 for the first time, and they are not even mentioned either in the text related to Figure 2 or in the caption of Figure 2.  The only mention of the Fourier transform in the main text is related to Figure 3, and it is just one sentence. Again in Figure 4, Fourier transforms are presented but not discussed in the text or in the figure caption. Either the authors believe these Fourier transform images convey important information and in this case, what they show should be discussed, or they don't, and in this case, maybe it is better to remove them or to place them in supplementary material.

3- Still about the Fourier transform patterns: Fourier transform patterns of sharp vertical lines are a single horizontal line that shows "interruptions".  On the Fourier transforms shown in this paper, we can see this "dotted" horizontal line, but there are also many vertical lines. Where do they come from? Actually, I used portions of the figures presented in the paper to perform the Fourier transforms myself, and did not get the vertical lines that can be seen on the patterns shown in inserts. How did you generate these Fourier transforms? Did you use a specific program or did you write the program yourself?

4- Legend of Figure 3d. PMMA patterns were obtained with a Zn mold, an Al mold, and a Ni mold. Why not place a picture of each case in Figure 3d? 

5- Lines 131 and 132: There is no need to mention EDS here, as you removed the EDS pictures from the paper.

Moderate editing of English is needed.

Author Response

[General review] The authors have implemented a number of modifications in the text, which improved the manuscript. Some points still remain unresolved:

[Our response] We greatly appreciate the reviewer's positive and thoughtful evaluation of our revised manuscript.

* [Comment 1], [Comment 2], and [Comment 3] are closely related.

[Comment 1] Figure 2, 3 and 4, Fourier transform patterns: These are still too small and too dark to read. Please rearrange the elements in these figures such that the Fourier transforms are larger and the reader can see the information these Fourier transforms convey.

[Comment 2] Fourier transform patterns: These Fourier transform patterns are not discussed in the main text. In fact, they are shown in Figure 2 for the first time, and they are not even mentioned either in the text related to Figure 2 or in the caption of Figure 2.  The only mention of the Fourier transform in the main text is related to Figure 3, and it is just one sentence. Again in Figure 4, Fourier transforms are presented but not discussed in the text or in the figure caption. Either the authors believe these Fourier transform images convey important information and in this case, what they show should be discussed, or they don't, and in this case, maybe it is better to remove them or to place them in supplementary material.

[Comment 3] Still about the Fourier transform patterns: Fourier transform patterns of sharp vertical lines are a single horizontal line that shows "interruptions".  On the Fourier transforms shown in this paper, we can see this "dotted" horizontal line, but there are also many vertical lines. Where do they come from? Actually, I used portions of the figures presented in the paper to perform the Fourier transforms myself, and did not get the vertical lines that can be seen on the patterns shown in inserts. How did you generate these Fourier transforms? Did you use a specific program or did you write the program yourself?

[Our response 1, 2, and 3] We appreciate the reviewer’s comment. We agree with the reviewer that the FFT patterns are unclear and unhelpful. In response to the reviewer’s comments, we deleted all fast Fourier Transform patterns in the revised manuscript.

[Modification of the manuscript]

on line 183 of the revised manuscript.

on line 204 of the revised manuscript.

“(c) The replica patterns from the surface-patterned metal master molds. The insets show photographic images of the replica pattern from the imprinted metal molds.  and fast Fourier transform patterns, indicating reliable pattern replication with high-resolution nanostructures.” on caption of the Figure 3.

The bottom-right inset images in Figure 4a show fast Fourier transform (FFT) patterns of the SEM images, suggesting well-arranged periodic line structures.” on line 211 of the revised manuscript.

on line 219 of the revised manuscript.

[Comment 4] Legend of Figure 3d. PMMA patterns were obtained with a Zn mold, an Al mold, and a Ni mold. Why not place a picture of each case in Figure 3d? 

[Our response 4] We greatly appreciate the reviewer’s comment. We agree with the reviewer’s suggestion to provide SEM images of all replicated PMMA patterns from the different metal molds. In response to the reviewer’s comment, we added cross-section view SEM images of the PMMA replica patterns obtained from the surface-patterned Zn, Al, and Ni molds as follws.

[Modification of the manuscript]

on line 204 of the revised manuscript.

“(d) Cross-section SEM images of replicated high-resolution PMMA patterns (line-width/ space-width: 250 nm) from the Zn, Al, and Ni molds. Scale bars, 1 µm.” on caption of the Figure 3.

[Comment 5] Lines 131 and 132: There is no need to mention EDS here, as you removed the EDS pictures from the paper.

[Our response 5] We really appreciate the reviewer’s helpful comment. In response to the reviewer’s suggestion, we removed the EDS data in the Materials and Methods section.

[Modification of the manuscript]

Energy-dispersive X-ray spectroscopy (EDS, INCA Energy X-MAX 50, Oxford Instrument) was used to analyze the elements of the imprinted metal samples.” on line 131 of the revised manuscript.
